# Participant acceptability of exercise in kidney disease (PACE-KD): a feasibility study protocol in renal transplant recipients

Nicolette C Bishop,[1] Roseanne Billany,[1,2] Alice C Smith[2,3]

[1]School of Sport, Exercise and Health Sciences, Loughborough University, Loughborough, UK
[2]John Walls Renal Unit, University Hospitals of Leicester NHS Trust, Leicester, UK
[3]Department of Infection, Immunity and Inflammation, University of Leicester, Leicester, UK

**Correspondence to**
Roseanne Billany;
r.billany@lboro.ac.uk

## ABSTRACT

**Introduction** Cardiovascular disease (CVD) is a major cause of mortality in renal transplant recipients (RTRs). General population risk scores for CVD underestimate the risk in patients with chronic kidney disease (CKD) suggesting additional non-traditional factors. Renal transplant recipients also exhibit elevated inflammation and impaired immune function. Exercise has a positive impact on these factors in patients with CKD but there is a lack of rigorous research in RTRs, particularly surrounding the feasibility and acceptability of high-intensity interval training (HIIT) versus moderate-intensity continuous training (MICT) in this population. This study aims to explore the feasibility of three different supervised aerobic exercise programmes in RTRs to guide the design of future large-scale efficacy studies.

**Methods and analysis** Renal transplant recipients will be randomised to HIIT A (16 min interval training with 4, 2 and 1 min intervals at 80%–90% of peak oxygen uptake ($VO_{2peak}$)), HIIT B (4×4 min interval training at 80%–90% $VO_{2peak}$) or MICT (~40 min cycling at 50%–60% $VO_{2peak}$) where they will undertake 24 supervised sessions (approximately thrice weekly over 8 weeks). Assessment visits will be at baseline, midtraining, immediate post-training and 3 months post-training. The study will evaluate the feasibility of recruitment, randomisation, retention, assessment procedures and the implementation of the interventions. A further qualitative sub-study QPACE-KD (Qualitative Participant Acceptability of Exercise in Kidney Disease) will explore patient experiences and perspectives through semistructured interviews and focus groups.

**Ethics and dissemination** All required ethical and regulatory approvals have been obtained. Findings will be disseminated through conference presentations, public platforms and academic publications.

**Trial registration number** Prospectively registered; ISRCTN17122775.

### Strengths and limitations of this study

► This study provides a multidisciplinary approach with a variety of quantitative and qualitative outcome measures which will contribute to future large-scale randomised control trial design with focus on negative cardiovascular disease outcomes in renal transplant recipients (RTRs).

► This study addresses the urgent need to determine the feasibility of differing aerobic exercise types in RTRs to inform future advice and recommendations.

► This study specifically contributes to the limited literature on the feasibility of high-intensity interval training in solid organ transplant patients.

► As a limitation, the sample size is limited due to the study being centred on feasibility in preparation for future efficacy trials.

## INTRODUCTION

Cardiovascular disease (CVD) is a major cause of morbidity and mortality after kidney transplantation and a key factor limiting graft survival.[1][2] Most recent data suggest that the proportion of mortalities in kidney transplant recipients attributed to CVD or stroke is 22%[3]; however, this figure does not emphasise the total impact on graft survival and quality of life of non-fatal events such as acute myocardial infarction, heart failure, stroke and cardiac arrhythmias.[2] Numbers of transplants per year are currently rising,[3] highlighting the need for cost-effective CVD prevention and risk factor management strategies for sustained patient health, graft survival and reduced healthcare burden.[4]

Traditional models of predicting CVD risk underestimate the risk in patients with chronic kidney disease (CKD) suggesting that kidney disease and transplantation introduce unique CVD risk factors not akin to the general population.[5] Immunosuppressive medication can exacerbate traditional risk factors such as poor lipid profile, elevated blood pressure and the incidence of new onset diabetes.[6][7] Weight gain, predominantly fat, is common in renal transplant recipients (RTRs) with 60% having a midarm fat area that is within the top 10% of values for the general population.[8] Unique haemodynamic challenges (anaemia, hypertension and volume expansion) in RTRs can accelerate cardiomyopathy in the absence

of the usual observation of concurrent ischaemic heart disease.[2] These haemodynamic stresses are a likely contributor to the paradoxically common vascular and systemic inflammation observed in RTRs.[9] Studies have suggested that elevations in novel biomarkers of inflammation are not just indicative but also contributory to CVD development.[10 11]

Empirical evidence suggests that physical activity has 'anti-inflammatory' consequences.[11] Recent exercise intervention studies in kidney patients who have not reached end-stage renal disease demonstrate reduced traditional and novel markers of CVD risk and improved physical function.[12–14] Similarly, dialysis patients have shown lowered markers of systemic inflammation[15] and improvements in arterial stiffness.[16] However, given its probable benefits, there is a shortfall in literature surrounding the role of physical activity in the management of CVD risk in RTRs.[17] With a precarious balance of immune function and non-traditional CVD risk factors, current exercise guidelines may not be appropriate for this population.

Moderate-intensity continuous training (MICT) such as 30–40 min brisk walking on 3 days of the week[12] or intradialytic cycling[15] have been used and shown to be beneficial in reducing markers of CVD risk in patients with kidney disease. There has been a recent abundance of literature surrounding high-intensity aerobic protocols, particularly high-intensity interval training (HIIT) which suggests a greater improvement in traditional and inflammatory markers of CVD risk in clinical and non-clinical populations when compared with MICT.[18] Although protocols differ, HIIT is characterised by short bursts of vigorous exercise (upwards of 80% maximal aerobic capacity) interspersed with periods of moderate intensity exercise or rest. In patients with CVD and heart transplant recipients, findings indicate that HIIT is well tolerated, safe and effective in improving CVD risk.[19–21] Most recently, HIIT has been shown to be effective in improving physical function, inflammation and quality of life in peritoneal dialysis patients[22]; however, the effectiveness of HIIT versus MICT in RTRs has not yet, to our knowledge, been evaluated. With a lower time commitment and the aforementioned superior benefits, HIIT has the potential to provide a creditable alternative to MICT. This type of exercise has also been perceived to be more enjoyable than MICT,[23] a key component for changing and maintaining physical activity levels in RTRs identified through qualitative methods.[24] However, our patients' views suggest that although HIIT may provide superior improvements of CVD risk factors with a shorter time commitment, the strenuous nature of the exercise might in itself present a barrier to participation in this population,[24] and MICT such as brisk walking, may be more acceptable and encourage sustained participation.

Considering both the development of effective and sustainable exercise guidelines for lowering the CVD risk in RTRs and producing programmes with a wide appeal and good adherence, this pilot research aims to generate substantial foundations towards understanding the feasibility and acceptability of differing aerobic exercise protocols in this unique population. The finding will contribute to the growing body of information not only in this population but in other chronic disease populations at risk of CVD and also determine the feasibility of proceeding to large and complex randomised control trials with primary outcomes focusing on the reduction of negative CVD outcomes.

## Objectives
### Primary objective
To evaluate the feasibility of a trial protocol involving three different supervised aerobic exercise programmes in renal transplant recipients.

### Secondary objectives
1. To assess participant experience and acceptability;
2. To explore potential for impact on physical function, body composition, habitual physical activity and non-invasive CVD risk factors;
3. To explore potential for impact on relevant blood, saliva and urinary markers of metabolic health, systemic inflammatory status and immune function;
4. To assess the tolerability, practicality and patient perceived usefulness of these assessments of cardiovascular disease risk;
5. To explore patient perception of their illness and symptoms, activity levels and quality of life.

## METHODS AND ANALYSIS
### PACE-KD trial design
The PACE-KD trial is a randomised, three-arm parallel-group study to determine the feasibility and acceptability of three different supervised aerobic exercise programmes. Figure 1 presents the study flow chart. There will be three parallel groups (exercise programmes): HIIT intervention A, HIIT intervention B and MICT (detailed below). Participants will be randomised to one of the three study arms and will complete 24 supervised sessions of the intervention (approximately three sessions per week over 8 weeks). After initial baseline testing, outcome measures will be attained at midtraining, immediate post-training and 3 months post-training. Additional acute outcome measures will be taken during one exercise session in the first 6 weeks and during one exercise session in the last 6 weeks of each exercise programme (see figure 2). Feasibility will be determined by assessing recruitment, compliance and completion. Acceptability and patient experience will be determined by a patient satisfaction questionnaire (PSQ) and an optional nested qualitative study (QPACE-KD) consisting of either one to one semistructured interviews or focus groups.

### Patient involvement
The study was designed in consultation with patient partners. A patient involvement (PPI) group has been convened (lead by JS) and will meet regularly with the rest of the research team to review progress and address any issues that arise. The PPI partners will particularly take part in the qualitative analysis and interpretation and dissemination of the results.

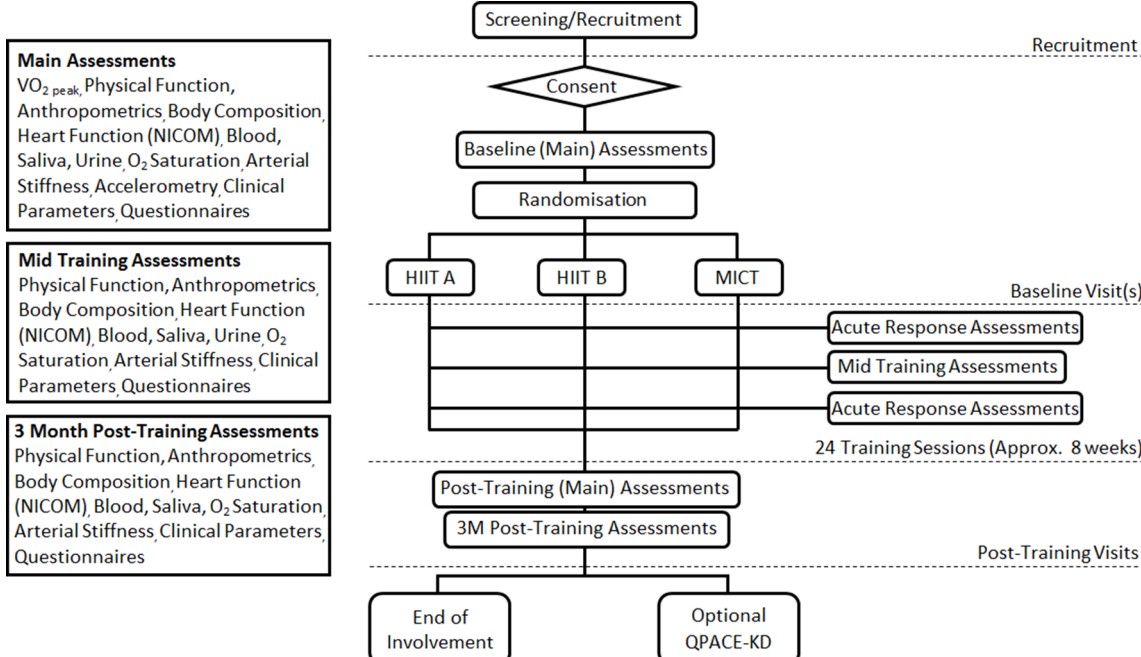

**Figure 1** Study flowchart: overall study diagram. HIIT, high-intensity interval training; MICT, moderate-intensity continuous training; NICOM, Non-Invasive Cardiac Output Monitor; VO$_2$, oxygen uptake.

## Participants and recruitment

A total of 24 participants with an established kidney transplant will be randomised (via computerised random number generator) to either HIIT intervention A, HIIT intervention B or MICT (see table 1 for detailed inclusion and exclusion criteria). Patients will be screened by their own clinician for eligibility to enter the study. Eligible patients will be approached during their routine clinical appointments and will be provided with verbal and written study information. Additionally, eligible patients who have given prior consent to be contacted regarding research opportunities will be contacted via the post. All patients will be given the opportunity to discuss the study in more detail and the opportunity to consider their participation (at least 48 hours).

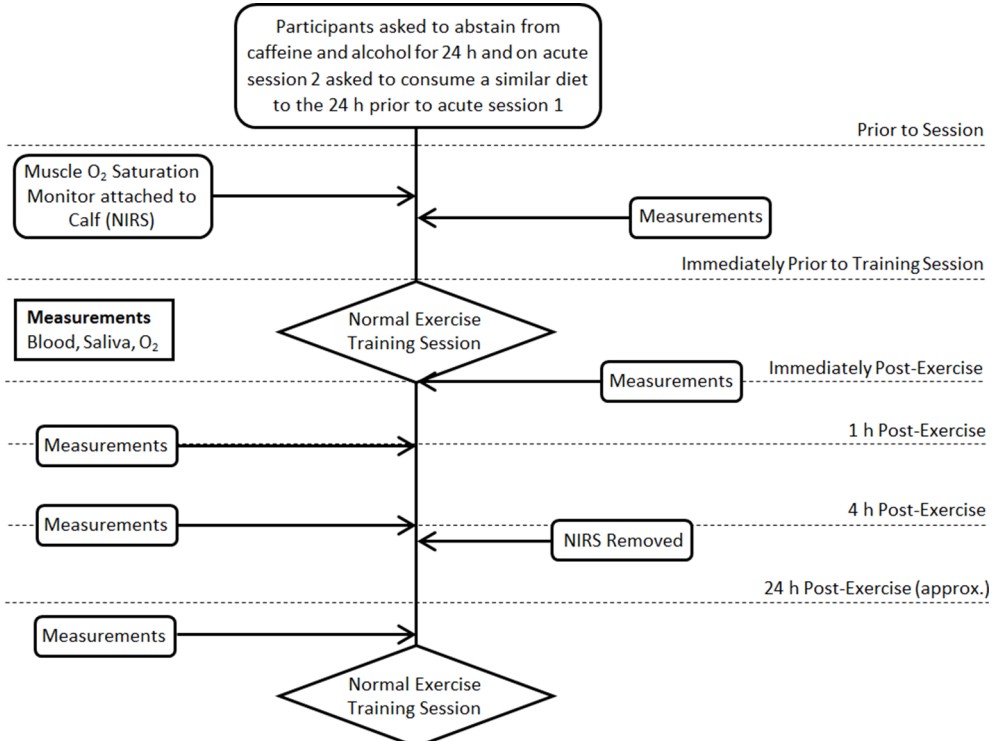

**Figure 2** Study flowchart: acute response sessions detailed diagram. NIRS, Near-Infrared Spectroscopy.

**Table 1** PACE-KD inclusion and exclusion criteria

| Inclusion criteria | Exclusion criteria |
| --- | --- |
| ► Diagnosed with chronic kidney disease and are a renal transplant recipient<br>► Male or female, aged 18 years or over<br>► Received renal transplant >12 weeks prior to entering the study<br>► Is willing and able to give informed consent for participation in the study | ► Aged under 18 years<br>► Female participants who are pregnant, lactating or planning pregnancy during the course of the study<br>► Scheduled elective surgery or other procedures requiring general anaesthesia during the study<br>► Any other significant disease or disorder*<br>► Inability to give informed consent or comply with testing and training protocol for any reason |

*That is, significant comorbidity including unstable hypertension, potentially lethal arrhythmia, myocardial infarction within 6 months, unstable angina, active liver disease, uncontrolled diabetes mellitus (HbA1c≥9%), advanced cerebral or peripheral vascular disease which, in the opinion of the patient's own clinician, may either put the patient at risk because of participation in the study or may influence the result of the study or the patient's ability to participate in the study.

### Interventions

Each exercise programme will consist of 24 sessions over approximately 8 weeks. Participants will be allowed to drink plain water during exercise sessions. The programmes will be based on the three protocols described below.

### HIIT programme A: (figure 3A)

Each session will consist of 16 min interval training with intervals of 4, 2 and 1 min duration at 80%, progressing to 90% oxygen uptake peak ($VO_{2\ peak}$) over the 8 weeks, separated by a 2 min active rest (~60% $VO_{2\ peak}$) giving a total exercise time of 30 min.

### HIIT programme B: (figure 3B)

Each supervised session will consist of 4×4 min of interval training at 80% $VO_{2\ peak}$, progressing to 90% $VO_{2\ peak}$ over the 8 weeks, separated by a 3 min active rest (~60% $VO_{2\ peak}$) and with a final 5 min recovery stage to ensure that overall work done is matched for both HIIT protocols, therefore also giving a total exercise time of 30 min.

### MICT exercise intervention

Each supervised session will consist of continuous brisk cycling for ~40 min with a target rating of perceived exertion (RPE) of 12–14 (somewhat hard). This is equivalent to 50%–60% $VO_{2\ peak}$ and will be matched to the HIIT protocols for total external work done based on calculations of percentage of peak watts obtained from the $VO_{2\ peak}$ test. In this way, for each individual the overall energy expenditure during the session would be the same regardless of the trial they are assigned to.

All exercise sessions will be preceded by a 5 min low intensity warm up and followed by a 10 min cool down.

### Study procedures

The study outcomes, measurement methods and assessment time periods are summarised in table 2, and a detailed overview of each measurement is provided at the end of this section. Data on recruitment, numbers of sessions attended and completed in accordance with the protocol and dropout will be recorded and continuously monitored.

### Baseline visit (visit 1)

At visit 1, informed consent will be obtained on arrival. Participants will perform a standard maximal exercise test ($VO_{2\ peak}$ test) in the presence of a cardiac nurse. This test will be preceded by a 12-lead ECG and subject to the confirmed absence of any contraindications.

Measurements of cardiac function, physical function, anthropometry, body composition and blood, urine and saliva samples will be collected. A survey pack will be completed and an accelerometer given to measure habitual physical activity over approximately 7 days.

### Training visits

Participants will attend 24 supervised exercise sessions (approximately three per week for 8 weeks) of either: HIIT A, HIIT B or MICT (as outlined above). At the beginning of each session, participants will be asked to complete an internally designed illness–symptom questionnaire. They will also be provided with a copy to complete over

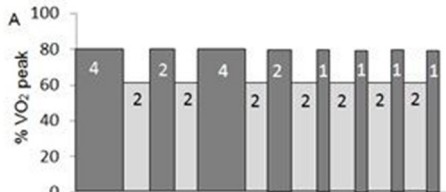 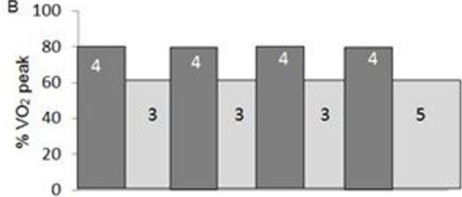

**Figure 3** Schematic diagram of HIIT regimens A and B. Each block shows stage duration in minutes. The final 5 min stage has been added to HIIT programme B to ensure the overall work done is matched for both protocols. HIIT, high-intensity interval training, VO₂, oxygen uptake.

**Table 2** PACE-KD outcomes, measurement methods and assessment time periods

| Outcome | Measurement method/tool | Time period |
|---|---|---|
| Feasibility | Recruitment, compliance, dropout and completion percentages | Continuous |
| VO$_{2peak}$ | CPET | Baseline, post-training |
| Physical function | Gait speed test | Baseline, midtraining, post-training, three months post-training |
|  | Sit to stand test | Baseline, midtraining, post-training, three months post-training |
|  | Plantar flexion strength (Fysiometer) | Baseline, midtraining, post-training, three months post-training |
| Body composition | BIA | Baseline, midtraining, post-training, three months post-training |
| Heart function | NICOM | Baseline, midtraining, post-training, three months post-training |
| Arterial stiffness | PWV (applanation tonometry) | Baseline, midtraining, post-training, three months post-training |
| SpO$_2$ | Pulse oximetry | Baseline, midtraining, post-training, three months post-training, acute response sessions |
| Habitual physical activity | Accelerometry | Baseline, post-training |
| Muscle O$_2$ saturation | NIRS | Acute response session |
| Urinary bacteriuria | Urine sample | Baseline, midtraining, post-training |
| Immune function | Blood and saliva samples | Baseline, midtraining, post-training, three months post-training, acute response sessions |
| Markers of inflammation, oxidative stress, CVD risk | Blood samples | Baseline, midtraining, post-training, three months post-training, acute response sessions |
| Renal profile | Blood samples | Baseline, post-training |
| Clinical information | Extraction from clinical notes | Baseline, midtraining, post-training, three months post-training |
| Renal symptoms | POS-S | Baseline, midtraining, post-training, three months post-training |
| Fatigue | Fatigue scale | Baseline, midtraining, post-training, three months post-training |
| Quality of life | EQ-5D | Baseline, midtraining, post-training, three months post-training |
| Illness perception | IPQ-R | Baseline, midtraining, post-training, three months post-training |
| Physical activity/functional capacity | DASI | Baseline, midtraining, post-training, three months post-training |
| Pain | BPI | Baseline, midtraining, post-training, three months post-training |
| Sleep quality | PSQI | Baseline, midtraining, post-training, three months post-training |
| Daytime sleepiness | ESS | Baseline, midtraining, post-training, three months post-training |
| Illness symptoms | PACE-KD Illness-Symptom Questionnaire | Before each exercise session and ×1 at the weekend between sessions |
| Tolerability, practicality, acceptability and perceived usefulness of the study measures | PSQ; Semi-structured interview or focus group | Post-study involvement; Post-study involvement through optional QPACE-KD nested qualitative study |

BIA, Bioelectrical Impedance Analysis; BPI, Brief Pain Inventory; CPET, Cardiopulmonary Exercise Test; CVD, Cardiovascular Disease; DASI, Duke Activity Status Index; EQ-5D, EuroQol 5 Dimension; ESS, Epworth Sleepiness Scale; IPQ-R, Illness Perception Questionnaire—Revised; NICOM, Non-Invasive Cardiac Output Monitor; NIRS, Near-Infrared Spectroscopy; POS-S, Renal Patient Outcome Scale; PSQ, Patient Satisfaction Questionnaire; PSQI, Pittsburgh Sleep Quality Index; SpO2, Peripheral Oxygen Saturation; VO$_{2peak}$, peak oxygen uptake.

the weekend between sessions. The questionnaire will provide data on illness symptoms particularly relating to upper respiratory, digestive and urinary tract.

### Acute response sessions

At one of the first six and at one of the final six exercise sessions, a resting blood sample will be collected immediately prior to the usual training session, immediately post-training session, 1 hour post-training session, 4 hour post-training session and immediately prior to the start of the next training session. Saliva samples will be taken in conjunction with each blood sample. Participants will be asked to abstain from caffeine and alcohol for 24 hours prior to these two acute testing sessions, and on acute testing session 2 asked to consume a similar diet to the 24 hours prior to acute session 1. During the training session, a muscle $O_2$ saturation monitor will be attached to the calf muscle (non-invasively) and will be removed following the 4 hour blood sample. Following the 1 hour samples, prior to the 4 hour sample, patients will be provided with light refreshments. They will be asked to refrain from consuming alcohol and caffeine and from partaking in strenuous exercise.

### Midtraining visit

The midtraining visit will take place between weeks 3 and 5. At this visit, the experimental tests described in the baseline visit will be repeated with the exception of the $VO_{2 peak}$ test and the accelerometer for habitual physical activity. The survey pack as described in the baseline visit will be administered, completed and collected during this visit.

### Immediate post-training visit

The immediate post-training visit will take place within 14 days of the final exercise session taking place. All experimental tests and the survey pack outlined in the baseline visit will be repeated. An accelerometer will be fitted as in the baseline visit and returned after 1 week.

### Three month post-training visit

The three month post-training visit will take place approximately three months following the final exercise session. The visit will be very similar to the midtraining visit. All experimental tests and the survey pack outlined in the aforementioned visit will be repeated with the exclusion of the $VO_{2 peak}$ test. There will be no more experimental measures after this visit.

An internally designed patient satisfaction questionnaire (PSQ) will be administered once at the end of study participation. The PSQ is designed to assess the acceptability of the study and experiences of the patients in each of the trial groups with particular focus on tolerability, practicality, enjoyment and perceived usefulness of the study measures and of the study as a whole. Participants who cease involvement in the study prior the final assessments will also be invited to complete the questionnaire. This will be optional.

## Study outcome measures

### $VO_{2 peak}$ test

Standard ramp protocol; increasing workload of approximately 1 W every 4 s (10–15 w/min) ensuring volitional exhaustion within 12–15 min[25] on a stationary electronically braked cycle ergometer (Lode Excalibur Sport, Groningen, Netherlands). Participants will be encouraged to cycle at a continuous cadence (~70 rpm). The highest oxygen uptake will be measured in this study ($VO_{2 peak}$) as true maximal $VO_2$ ($VO_{2 max}$) is less commonly achieved in deconditioned and/or clinical patients. Results of this test will be used to determine training workloads.

### Clinical information

Clinical information will be extracted from the medical notes including: age, gender, ethnicity, primary cause of renal failure, transplant type, time since transplant, dialysis duration, comorbidities, blood/urine results, current medication and smoking habits. This information will be used to primarily account for cofounding variables and analyses of differences and similarities between intervention arm groups.

### Anthropometric measures

Anthropometric measures of height, weight and waist and hip circumference will be attained in accordance with standard protocols.[26]

### Body composition

Bioelectrical impedance analysis performed on an InBody analyser (InBody 370, Chicago, Illinois, USA) will be used to estimate body composition (eg, body fat percentage and fat-free mass). This is a painless, non-invasive method[27 28] previously validated for use in patients with CKD .[29]

### Cardiovascular condition and function

A non-invasive cardiac output monitor (Cheetah Medical, Maidenhead, UK) will be used to assess heart function using cardiac bioreactance analysis. This method is safe, quick and validated[30] for assessing haemodynamic parameters including cardiac output (heart rate x stroke volume) and total peripheral resistance.

### Pulse wave analysis

Pulse wave analysis will be measured non-invasively to determine arterial stiffness using applanation tonometry methods (Sphygmocor, AtCor Medical, West Ryde, New South Wales, Australia). Arterial pressure waveforms are recorded superficially at the radial artery by an experienced laboratory technician by applanating the artery with a high-fidelity hand-held tonometer at the site of maximal pulsation. Consistent and reproducible waveforms are recorded for 20 s. This is a safe procedure and a measure of this type has been performed in patients with CKD previously.[13 14]

### Physical function

Physical function will be assessed using three tests. The sit-to-stand (STS) test will be used to ascertain functional

ability and has been widely used in the CKD population.[31–33] Participants will be asked to sit on a hard upright chair with their feet flat on the floor and knees bent at 90°. One cycle is considered complete when the participant stands up and sits down again to the starting position (without using their hands). There are two tests within the STS:

► STS5: The time taken for patients to stand up from a seated position and sit back down again five times.
► STS60: The number of STS cycles achieved in 60 s and is a surrogate measure of muscular endurance.

The 4 m walk test will be used to assess gait speed. Participants will be asked to walk 4 m at their usual pace for one practice and two timed trials. The better score (in seconds) on the timed trials will be recorded.

Isometric plantar flexion strength (gastrocnemius and soleus strength) will be assessed using a modified Wii Fit board (Nintendo, Kyoto, Japan). Participants will sit with their legs bent to 90° with their feet flat on the board. A strap will be placed around the board and over the knees. Participants will then be asked to raise their heels off the board causing a downward force to be exerted on the board.

### Muscle oxygen saturation

Prior to the acute testing sessions a small non-invasive muscle $O_2$ saturation device (BSXInsight, BSXAthletics, Texas, USA) will be placed onto the calf muscle and secured with a small bandage. The device will be removed following the 4hour sample collection. This will measure muscle $O_2$ saturation using NIRS and has been used in a variety of diseases previously.[34]

### Saliva sampling and storage

Saliva samples will be collected into sterile plastic containers. Participants will swallow to empty the mouth, then open and hold the container themselves before performing a passive dribble of saliva collected under the tongue over the next 2 min. Following centrifugation, the supernatant will be aliquoted and frozen for future analysis, primarily for secretory immunoglobulin A to investigate mucosal immunity.

### Urine sampling and storage

A urine sample will be requested during the baseline visit. This sample will be analysed for urinary protein excretion, bacteriuria and urinary tract infection susceptibility.

### Venous blood sampling and analysis

Venous blood will be collected using venepuncture of the antecubital vein and prepared appropriately for the following analysis:

► A cytometric bead array technique will be used to allow bulk analysis of a panel of proinflammatory and anti-inflammatory cytokines, including but not limited to interleukin (IL)-1, IL-2, IL-6, IL-10, tumour necrosis factor-alpha and interferon gamma.
► Other markers of inflammation, C-reactive protein (CRP) and oxidative stress, malondialdehyde (MDA) will be assessed by ELISA.
► Phenotyping of microparticles using flow cytometry.
► Immune cell subsets using flow cytometry.
► Full blood count and renal profile analysis, which includes estimated glomerular filtration rate, urea, bicarbonate, creatinine, sodium, potassium and phosphate measures.

### Questionnaires

Participants will be asked to complete a survey pack containing the following: (1) Renal Patient Outcome Scale; a validated questionnaire that measures the presence and severity of disease related symptoms with the addition of a transplant specific question set.[35] (2) Fatigue Scale; an internally designed Likert scale on feelings of tiredness. (3) EuroQol; a short widely used quality of life questionnaire.[36] (4) Illness Perception Questionnaire—Revised; an 84-item questionnaire which is used to assess the principle components of illness representations.[37] This questionnaire has been tailored for patients with chronic kidney disease and will assess their perceptions of physical activity as treatment. (5) Duke Activity Status Index; a 12-item questionnaire that uses self-reported physical work capacity to estimate peak metabolic equivalents and has been shown to be a valid measurement of functional capacity.[38] (6) Brief Pain Inventory—Short Form; a nine-item self-administered questionnaire used to evaluate the severity of a patient's pain and the impact of this pain on the patient's daily functioning.[39] (7) The Pittsburgh Sleep Quality Index; a self-report questionnaire that assesses sleep quality over a onemonth time interval[40] (8) Epworth Sleepiness Scale (ESS); the ESS measures a person's general level of daytime sleepiness or their average sleep propensity in daily life.[41]

### Accelerometry

An ActiGraph wGT3X-BT accelerometer (Actigraph, Pensacola, Florida, USA) will be provided to wear on the waist which will record habitual physical activity (returned after 1 week).

### Nested qualitative study (QPACE-KD)

Six participants from each intervention arm will be invited to an audiorecorded focus group or semistructured interview to discuss their experiences of the study and the intervention in more detail and to gather suggestions for relevant and acceptable design of future randomised controlled trials. The qualitative phase will be conducted by a practised researcher uninvolved with the aforementioned study protocol to encourage freedom of expression. Both one-to-one semistructured interviews and focus groups will be used to provide different but complementary qualitative information. The interviews will ascertain individual perspectives and feelings about the intervention and outcome protocol in the light of personal and individual experiences of having completed the feasibility study. On the other hand, focus groups will elicit perceptions which are particularly useful when there is some

consensus and meaning about the existence of a community or shared voice.

The QPACE-KD study is an optional addition to PACE-KD. The QPACE-KD protocol will be explained to the patient, and they will be given ample opportunity to ask questions and at least 48 hours to decide whether or not to participate. If they agree, separate QPACE-KD written informed consent will be attained before the interview or focus group takes place. During the interviews and focus groups, the facilitator will introduce the process, explaining the background and reasons for the study, how the interview will proceed and details of audio recording and note taking. Participants will be reassured about the preservation of their anonymity and confidentiality as well as given the opportunity to ask questions.

Transcripts will only include non-identifying information. Individual's names and personal details will not be included in the completed transcripts. Similarly, any written notes taken during the interviews will not include any identifiable personal data.

### Data collection and management

Data at all time points will be collected in case report forms by the trial team. All data will be entered into a secure database and will only be accessible on password-protected computers at University Hospitals of Leicester NHS Trust (UHL), University of Leicester and Loughborough University by relevant members of the study team. No identifying information will be kept in electronic form. All source data and original participant identities will be kept in a locked office in the trial site file only at UHL.

### Sample size

As the trial has a primary outcome of feasibility the study sample size is pragmatic and a power calculation is neither relevant nor possible. Data collected on secondary outcome measures will provide data on which sample size calculations can be performed in future well-informed randomised controlled trials. We have chosen specific and relevant secondary outcome variables that may be suitable future primary outcomes.

### Data analysis

The primary outcome will assess the feasibility of study components such as recruitment, randomisation, retention, assessment procedures and the implementation of the interventions. These will be quantified by analysing factors such as but not limited to: numbers screened, numbers enrolled, proportion of eligible patients enrolled, intervention-specific retention rates, rates of adherence to protocol per intervention and proportion of training sessions and assessment sessions completed. The criteria for determining study and intervention feasibility will be as follows: Compliance: attending >70% of sessions and retention/completion:>80% of participants.

Due to the small sample size, complex statistical analysis will not be performed on the secondary outcomes; however, we will use simple analysis (eg, t-test, linear regression) to investigate statistical differences for exploratory information only.

Interview data will be analysed according to the principles of interpretive thematic analysis using a framework approach[42] to explore themes emerging from patient journeys through and experiences of the interventions and outcome measures. The whole research team, including patient partners, will be involved in qualitative analysis to avoid individual bias.

## ETHICS AND DISSEMINATION
### Ethical issues

The University Hospitals of Leicester NHS Trust agreed to act as sponsor for this study on 31 October 2016 (EDGE 88714). The protocol was reviewed by the East Midlands-Nottingham Research Ethics Committee (REC) and was given a favourable opinion (REC ref 16/EM/0482) on 04 January 2017. Health Research Authority regulatory approval was given on 27 January 2017, and the study was adopted on the NIHR portfolio on 12 January 2017. Local governance approval was granted by UHL Research and Innovation on 02 March 2017. Steps have been taken when designing this protocol to minimise the ethical implications and ensure patient welfare. The study will comply with the International Conference for Harmonisation of Good Clinical Practice guidelines and the Research Governance Framework for Health and Social Care.

### Dissemination plan and impact

On completion, the results of this study will be published in peer-reviewed journals and presented at national and international conferences. Participant level data will be available following publication of results on request. Results will also be disseminated to the patient and public community via social media and newsletter articles and presentations at patient conferences and forums, lead by the patient partners.

It is anticipated that the results of this study will inform future design of larger randomised controlled trials in this subject area and contribute to future specific physical activity guidelines in this population.

## ACKNOWLEDGEMENTS
We would like to thank Dr Clare Stevinson (Loughborough University), Dr James O Burton (University Hospitals of Leicester NHS Trust) and Mr John Savage for their valuable work towards the development of the funding application for this study.

**Twitter** @RBillany

**Contributors** ACS is the chief investigator for this trial. ACS and NCB lead on study design and protocol preparation. RB wrote this manuscript and contributed to protocol preparation, ethics application and document preparation. All authors reviewed and approved the final version of this manuscript.

**Funding**  This study is jointly funded by Heart Research UK grant number RG2650/15/18, the Stoneygate Trust and the Engineering and Physical Sciences Research Council (EPSRC) Antimicrobial Resistance (AMR) grant. The research was supported by the National Institute for Health Research (NIHR) Leicester Biomedical Research Centre. The views expressed are those of the authors and not necessarily those of the NHS, the NIHR or the Department of Health.

**Competing interests**  None declared.

**Ethics approval**  East Midlands Nottingham REC - 16/EM/0482 - 216043.

**Provenance and peer review**  Not commissioned; externally peer reviewed.

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
