## [Reviewer comments · BMJ Open]

ARTICLE DETAILS

TITLE (PROVISIONAL)	Participant Acceptability of Exercise in Kidney Disease (PACE-KD): A feasibility study protocol in renal transplant recipients
AUTHORS	Bishop, Nicolette; Billany, Roseanne; Smith, Alice

VERSION 1 - REVIEW

REVIEWER	Zeynep Kendi Celebi Batman State Hospital Batman, Turkey
REVIEW RETURNED	15-May-2017

GENERAL COMMENTS	A good designed study, it will answer the questions about renal transplant recipients's exercise capacity and CVD risk
--

REVIEWER	Marcelo Santos Sampaio Allegheny Health Network, Pittsburgh, USA
REVIEW RETURNED	21-May-2017

GENERAL COMMENTS	This study proposal from Bishop at al intent to examine the physical tolerance of transplant recipients to different intensity exercise regimens. The protocol was well written and outcomes were well defined. However, the protocol may be too extensive to answer the primary outcome, and conclusions on secondary outcomes may not be possible. I have a few suggestions/comments regarding the trial design to the authors. 1. Authors have mentioned that more intense regimens were tolerated by ESRD and in PD patients in previous trials. Transplant patients are in general a selected population from the HD/PD/ESRD population with fewer or less severe CVD disease. The chance of tx patient not being able to tolerate exercise is small, except if a high-risk CV recipients were selected which is not the case given the proposed exclusion criteria. Having said that, to achieve the primary objective authors may not need such an extensive protocol. The small number of patients (24) may be also a problem to equilibrate baseline characteristics and account for the dropouts for noncompliance with the protocol. To achieve the main outcome, each patient could be submitted to one week of each physical regimen as a crossover study and compare the performances. This would be faster, cheaper and N would be 24 as opposed to at least 8 in each group. The secondary outcomes would be the subject of a second study. This time more comprehensive and with a well-defined power analysis.
---

	2. Noncompliance with the study protocol has to be better defined in the exclusion criteria section. 3. As the study is designed the data may not be conclusive to the secondary outcomes given the small number of patients per group. Authors have mentioned that data generated to secondary outcomes will be used to define power analysis to a subsequent trial. The power analysis of the second study can be inferred from the similar studies done in ESRD in or not in dialysis. 3. other concern is that including recipients early on after transplant, starting with 3 months after transplant, would be too early. These patients are still under a reasonable risk of infection and rejection complications and unstable kidney function. These complications may interfere with the participation of patients in the exercise activities. They are also exposed to higher doses of immunosuppression which may be associated with more hypertension and etc... Maybe the ideal time to enroll patients would be after 6 months or a year after transplant. 4. There is no specification how cardiovascular function will be measured, by ECHO, stress test, functional scale. Since patients that receive a transplant have CV function fairly studied in the pre-transplant these exams/tests can be used to define baseline CV function. 5. Authors should mention how groups performances will be statistically compared. Given the small number of patients in each group for the secondary outcomes adjustments for different patient characteristics, drug use and etc... may be needed.
--	--

VERSION 1 – AUTHOR RESPONSE

Reviewer: 1

Zeynep Kendi Celebi

Batman State Hospital Batman, Turkey

Please state any competing interests or state 'None declared': None declared

Please leave your comments for the authors below

A good designed study, it will answer the questions about renal transplant recipients's exercise capacity and CVD risk

We would like to thank reviewer 1 for the positive comments. We are pleased that you thought that the study was well designed.

Reviewer: 2

Marcelo Santos Sampaio

Allegheny Health Network, Pittsburgh, USA

Please state any competing interests or state 'None declared': None declared

Please leave your comments for the authors below

This study proposal from Bishop at al intent to examine the physical tolerance of transplant recipients to different intensity exercise regimens. The protocol was well written and outcomes were well defined. However, the protocol may be too extensive to answer the primary outcome, and conclusions

on secondary outcomes may not be possible.

We would like to thank the reviewer 2 for their positive comments. However, the primary outcome is “To evaluate the feasibility of a trial protocol involving three different supervised aerobic exercise programmes in renal transplant recipients”. This will be quantified via recruitment, retention and completion data – so we are confident we will be able to measure these. As regards secondary outcomes, we have not stated that we will be able to make definite conclusions on these as the study is designed to test feasibility, not the efficacy of the intervention. Proper feasibility testing is a vital step of clinical trial design that is often overlooked. Feasibility testing should include evaluation of all elements of an eventual full trial, including recruitment process, and tolerability, relevance and usefulness of outcome measures as well as the intervention itself, hence the reason we have included them. We do not expect to make firm conclusions of efficacy from the outcome measures but we will be able to evaluate their acceptability and usefulness, and the qualitative substudy will be informative about the patient perspective of their relevance and the burden involved in undertaking them

I have a few suggestions/comments regarding the trial design to the authors.

1. Authors have mentioned that more intense regimens were tolerated by ESRD and in PD patients in previous trials. Transplant patients are in general a selected population from the HD/PD/ESRD population with fewer or less severe CVD disease. The chance of tx patient not being able to tolerate exercise is small, except if a high-risk CV recipients were selected which is not the case given the proposed exclusion criteria. Having said that, to achieve the primary objective authors may not need such an extensive protocol. The small number of patients (24) may be also a problem to equilibrate baseline characteristics and account for the dropouts for noncompliance with the protocol. To achieve the main outcome, each patient could be submitted to one week of each physical regimen as a crossover study and compare the performances. This would be faster, cheaper and N would be 24 as opposed to at least 8 in each group. The secondary outcomes would be the subject of a second study. This time more comprehensive and with a well-defined power analysis.

We thank reviewer 2 for these comments, however the study is designed to provide vital information for the future design of a robust RCT. We want to see how each arm recruits, retains and is completed so we can use this to design the future RCT which will assess efficacy. For the RCT we know a longer trial will be needed to truly assess the physiological benefits of the exercise, but at this stage we need to see if the exercise itself is acceptable on a number of fronts from both the views on the protocol, the time commitment required etc as well as the outcome measures. We cannot assess this in a one week study, and would still require doing the second study to be able to design the RCT. The point of the study is to assess the feasibility of these exercise protocols – if they cannot be completed or are not viewed as acceptable then this gives us vital information about their feasibility. With regard to sentence 1 of the reviewer comments, there is one study outlined in the protocol which does show some beneficial effects of HIIT in PD patients. However, this protocol is less intense than the proposed study and transplant recipients have unique CVD risk factors not akin to the general population and other disease states. There are also other factors to consider in this population such as the immunosuppressive medication (and its potential interaction with exercise) which you would not see in non-transplant CKD patients. Transplant patients have a unique perspective on exercise regimes as they are concerned to protect their precious graft but also want to live life to the full and make the most of the gift. The qualitative substudy will explore this and to gain such insight into the lived experience of participating in HIIT we need to give the participants a reasonable training period. We chose 8 weeks as we believe this will be practical but long enough to become accustomed to the exercise protocol (which can take longer than 1 week).

2. Noncompliance with the study protocol has to be better defined in the exclusion criteria section.

The exclusion criteria section is solely for the purpose of eligibility for recruitment into the study. We do not intend to exclude participants for non-compliance (other than voluntary drop-out), as the whole purpose is to elucidate the extent to which these exercise regimes are tolerable and acceptable. However we agree that noncompliance with the study protocol has not been stated in the manuscript. As this is a feasibility study assessing the feasibility of the study protocol, noncompliance with the study protocol will form a useful part of the outcome measures. Under the 'Data Analysis' section heading on page 11 we have added criteria for how we will determine study and intervention feasibility. You will see that compliance is now stated as attending >70% of sessions.

3. As the study is designed the data may not be conclusive to the secondary outcomes given the small number of patients per group. Authors have mentioned that data generated to secondary outcomes will be used to define power analysis to a subsequent trial. The power analysis of the second study can be inferred from the similar studies done in ESRD in or not in dialysis.

Unfortunately there are not similar studies in this very specific population who are on immunosuppressive medication and the interactions with exercise (and its known effects on immune function) are not known, which is why we need to use the secondary outcomes from this study to help us define n for a future RCT. This population are not directly comparable with either ESRD on dialysis or non-dialysis populations.

3. other concern is that including recipients early on after transplant, starting with 3 months after transplant, would be too early. These patients are still under a reasonable risk of infection and rejection complications and unstable kidney function. These complications may interfere with the participation of patients in the exercise activities. They are also exposed to higher doses of immunosuppression which may be associated with more hypertension and etc... Maybe the ideal time to enrol patients would be after 6 months or a year after transplant.

We accept this point and some of these issues will be covered by our exclusion criteria. In particular exclusion criteria from Table 1 on page 5: 'significant co-morbidity including unstable hypertension, potentially lethal arrhythmia, myocardial infarction within 6 months, unstable angina, active liver disease, uncontrolled diabetes mellitus (HbA1c \geq 9%), advanced cerebral or peripheral vascular disease which, in the opinion of the patient's own clinician, may either put the patient at risk because of participation in the study, or may influence the result of the study, or the patient's ability to participate in the study.'

Again the feasibility nature of the study will help to determine the ideal time to enrol patients.

4. There is no specification how cardiovascular function will be measured, by ECHO, stress test, functional scale. Since patients that receive a transplant have CV function fairly studied in the pre-transplant these exams/tests can be used to define baseline CV function.

We feel that we have covered the details of how CV function will be measured within the original manuscript. From page 6 under the heading 'Baseline Visit (Visit 1)': 'Participants will perform a standard maximal exercise test (VO₂ peak test) in the presence of a cardiac nurse. This test will be preceded by a 12-lead electrocardiogram (ECG) and subject to the confirmed absence of any contraindications.'

We will also measure 'Cardiovascular Condition and Function' using NICOM which is described on page 8 under the heading 'Study Outcome Measures' as follows: 'A Non-Invasive Cardiac Output Monitor (NICOM; Cheetah Medical, Maidenhead, UK) will be used to assess heart function using cardiac bioreactance analysis. This method is safe, quick and validated for assessing haemodynamic parameters including cardiac output (heart rate x stroke volume) and total peripheral resistance.'

5. Authors should mention how groups performances will be statistically compared. Given the small

number of patients in each group for the secondary outcomes adjustments for different patient characteristics, drug use and etc... may be needed.

We would agree that this information should usually be included in the statistical analysis section, however due to the smaller sample size and the study not being focussed on efficacy, we feel that it is not appropriate to use complex statistical analysis on the secondary outcome measures. We will use these outcomes to perform power calculations for a future larger RCT. These outcome measures are also included in order to devise an appropriate set of measures for a future RCT from the feedback ascertained from participants. We have however, now included details of some simpler analysis that we will perform on the secondary outcomes to provide exploratory information. This can be found on page 11 under the 'Data Analysis' heading (the second paragraph).